# Low Levels of Hive Stress Are Associated with Decreased Honey Activity and Changes to the Gut Microbiome of Resident Honey Bees

Kenya E. Fernandes,[a] Bridie Stanfield,[a] Elizabeth A. Frost,[b,c] Erin R. Shanahan,[a,d] Daniel Susantio,[a] Andrew Z. Dong,[a] Trong D. Tran,[e] Nural N. Cokcetin,[a,f] Dee A. Carter[a,g]

aSchool of Life and Environmental Sciences, University of Sydney, Sydney, New South Wales, Australia
bABGU, A Joint Venture of NSW Department of Primary Industries and University of New England, Armidale, New South Wales, Australia
cNSW Department of Primary Industries, Paterson, New South Wales, Australia
dCharles Perkins Centre, University of Sydney, Sydney, New South Wales, Australia
eSchool of Science, Technology and Engineering, University of the Sunshine Coast, Maroochydore, Queensland, Australia
fAustralian Institute for Microbiology and Infection, University of Technology, Sydney, New South Wales, Australia
gSydney Institute for Infectious Diseases, University of Sydney, Sydney, New South Wales, Australia

**ABSTRACT** Honey bees (*Apis mellifera*) face increasing threats to their health, particularly from the degradation of floral resources and chronic pesticide exposure. The properties of honey and the bee gut microbiome are known to both affect and be affected by bee health. Using samples from healthy hives and hives showing signs of stress from a single apiary with access to the same floral resources, we profiled the antimicrobial activity and chemical properties of honey and determined the bacterial and fungal microbiome of the bee gut and the hive environment. We found honey from healthy hives was significantly more active than honey from stressed hives, with increased phenolics and antioxidant content linked to higher antimicrobial activity. The bacterial microbiome was more diverse in stressed hives, suggesting they may have less capacity to exclude potential pathogens. Finally, bees from healthy and stressed hives had significant differences in core and opportunistically pathogenic taxa in gut samples. Our results emphasize the need for understanding and proactively managing bee health.

**IMPORTANCE** Honey bees serve as pollinators for many plants and crops worldwide and produce valuable hive products such as honey and wax. Various sources of stress can disrupt honey bee colonies, affecting their health and productivity. Growing evidence suggests that honey is vitally important to hive functioning and overall health. In this study, we determined the antimicrobial activity and chemical properties of honey from healthy hives and hives showing signs of stress, finding that honey from healthy hives was significantly more antimicrobial, with increased phenolics and antioxidant content. We next profiled the bacterial and fungal microbiome of the bee gut and the hive environment, finding significant differences between healthy and stressed hives. Our results underscore the need for greater understanding in this area, as we found even apparently minor stress can have implications for overall hive fitness as well as the economic potential of hive products.

**KEYWORDS** antimicrobial honey, hive health, hive stress, honey bee, honey bee ecology, honey bee gut microbiome, medicinal honey

Honey bees (*Apis mellifera*) are integral to global biodiversity, food security, and the economy, playing a critical role as pollinators for many of the world's plants and crops, and producing economically important products such as honey and wax. Bee

Address correspondence to Dee A. Carter, dee.carter@sydney.edu.au.

The authors declare no conflict of interest.

colonies in Australia and around the world are being increasingly exposed to sources of stress associated with human activity, including degraded ecosystems and the expansion of agricultural monocultures resulting in nutritional stress (1) and pesticide exposure and contaminated landscapes resulting in behavioral changes (2) and increased pathogen susceptibility (3). While major stresses can rapidly and irreversibly disrupt a hive, minor stressors can also accumulate over time and cause a chronic weakening of colonies leading to issues such as low hive population, decreased food stores, loss of the queen, and increased pathogen susceptibility (4).

Growing evidence suggests that honey is vitally important to hive function and overall health. Beyond being a source of energy and nutrition for resident bees (5), honey protects the hive from microbial overgrowth and detoxifies nectar (6), and it enhances bee longevity (7), toxin tolerance (8), cold tolerance (9), and immune function (10). As a natural product, the antimicrobial properties of honey are highly variable and influenced by a wide variety of factors (11). While the antimicrobial properties of honey are well-established in the context of human health (12), fewer studies have connected this to bee health. Honey activity is also almost always studied in the context of floral source though many of its antimicrobial components originate from the bees that produce it, including antimicrobial peptides (13, 14) and glucose oxidase (15).

Key to bee health is the gut microbiome, which performs many beneficial functions, including aiding digestion (16, 17), protecting against the effects of chronic pesticide exposure (18), and priming the innate immune system against pathogens (19). The bacterial microbiome in the bee gut comprises conserved core phylotypes present in all adult bees (20), which are influenced by location, season, and environmental landscape (21–23). Composition varies along the digestive tract, from the crop with limited bacterial species that are associated with exposure to nectar and the hive environment, through the midgut with an intermediate composition, to the hindgut with the highest bacterial abundance composed primarily of the core taxa. The bee gut also contains a smaller fungal community (24), but this is less well understood. Under stressful conditions, bees are more susceptible to microbiome destabilization, which in turn has detrimental effects on their health (25). In addition to bees, the hive itself has its own microbiota, including in hive products such as bee bread and honey, and in the general hive environment.

Given the importance of honey and the gut microbiota in supporting bee health as well as the ecological and economic imperatives to understand and protect bees from hive stress, this study aimed to investigate the link between minor hive stress, honey activity, and the hive and bee gut microbiome. Samples were taken from three healthy hives and three hives exhibiting signs of stress from a single apiary in Paterson, New South Wales. The antimicrobial activity of honey samples was tested against a panel of bacterial and fungal pathogens, and chemical properties linked to antimicrobial activity were assayed, including color, hydrogen peroxide production, phenolics content, and antioxidant content. Bacterial and fungal microbiomes were profiled for gut sections from forager bees and swabs from hive components. We found hive health to be associated with significant changes in the antimicrobial activity and chemical composition of honey, and with differences in the bacterial and fungal microbiome of the gut and hive environment.

## RESULTS

**Healthy hives produce honey with significantly higher activity than stressed hives.** The antimicrobial activity of honey samples from each hive across the three time points was tested against a range of pathogenic microbes, including bacteria, yeasts, and molds in order to determine whether there were differences in activity and if these aligned with hive health. Both total activity, a test of the honey's overall activity, including hydrogen peroxide-based action, and nonperoxide activity, a test of the honey's remaining activity after catalase has been used to abolish the action of hydrogen peroxide, were determined. Individual values for each honey sample as well as artificial and control honeys are shown in Table S2. Fig. 1A and B compares the average

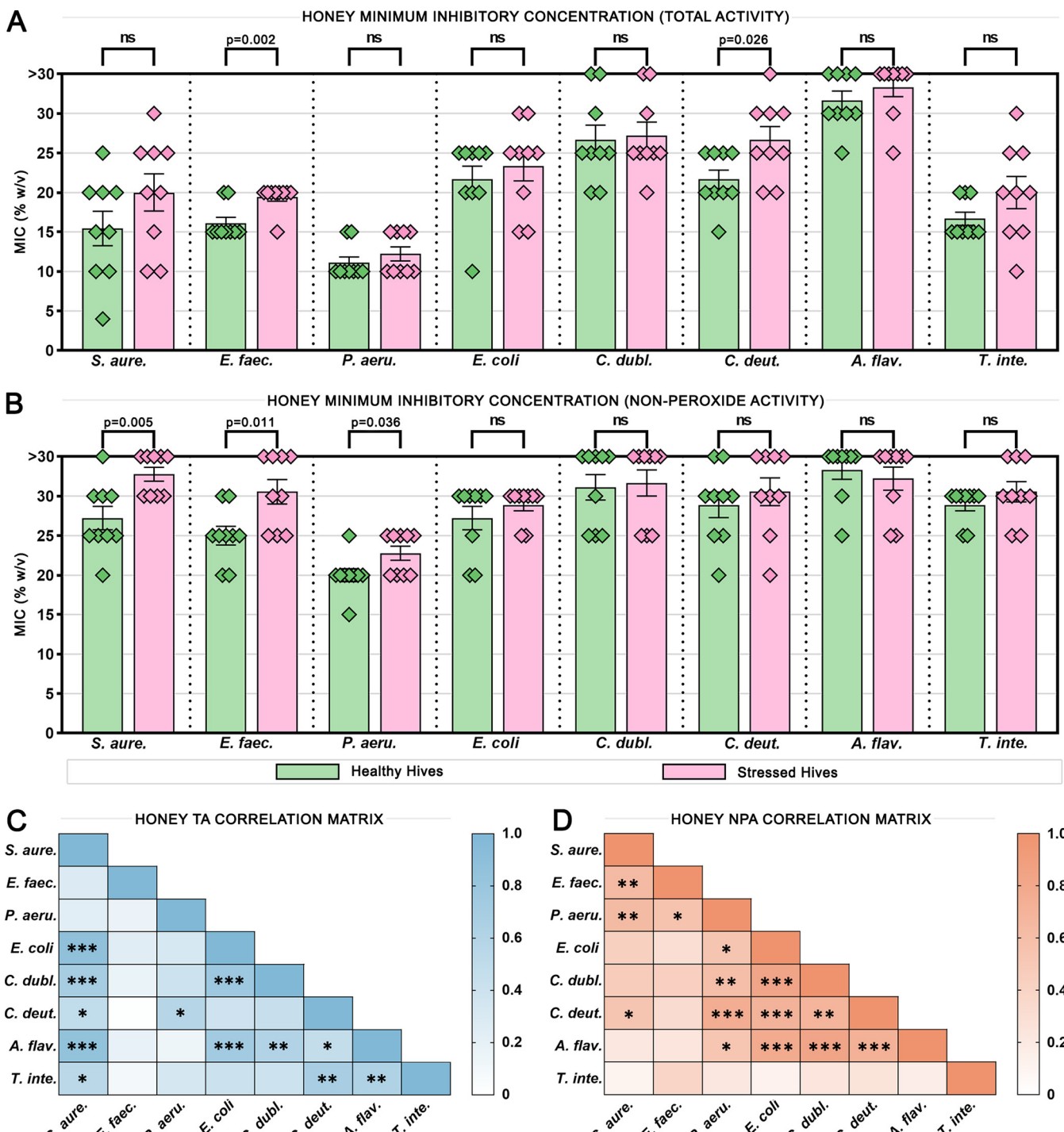

**FIG 1** Antimicrobial activity of honey samples against bacteria and fungi. (A) Total activity and (B) nonperoxide activity MICs of honey samples from healthy and stressed hives across three time points against bacterial pathogens *Staphylococcus aureus*, *Enterococcus faecalis*, *Pseudomonas aeruginosa*, and *Escherichia coli* and fungal pathogens *Candida dubliniensis*, *Cryptococcus deuterogattii*, *Aspergillus flavus*, and *Trichophyton interdigitale*. The % (wt/vol) refers to the concentration of honey diluted in sterile water (for total activity) or catalase solution (for nonperoxide activity). Columns represent the mean while diamonds show the values of individual honey samples. Error bars show SEM. (C) Total activity (TA) and (D) nonperoxide activity (NPA) correlation matrices showing how closely the antimicrobial activity of honey samples aligns between different species. All correlations were positive; colored scalebars indicate level of correlation. *, $P \leq 0.05$, **, $P \leq 0.01$, ***, $P \leq 0.001$.

activity of honey samples from healthy versus stressed hives. Honeys from healthy hives in general had lower MICs than honeys from stressed hives and were significantly more active against *E. faecalis* (3% [wt/vol] lower; $P = 0.002$) and *C. deuterogattii* (5% [wt/vol] lower; $P = 0.026$) for total activity, and against *S. aureus* (6% [wt/vol] lower; $P = 0.005$),

*E. faecalis* (6% [wt/vol] lower; $P = 0.001$), and *P. aeruginosa* (3% [wt/vol] lower; $P = 0.036$) for nonperoxide activity.

The MIC for artificial honey, a sugar solution used as a control for the osmolarity of honey, was >30% (wt/vol) for all species tested except the Gram-negative bacterium *P. aeruginosa*, which displayed increased susceptibility to sugar with an MIC of 25% (wt/vol) (Table S2). *P. aeruginosa* was also the most susceptible microbe to total honey activity (on average 12% [wt/vol]), followed by Gram-positive bacteria *S. aureus* and *E. faecalis* and the fungal dermatophyte *T. interdigitale* (18% [wt/vol]), Gram-negative bacterium *E. coli* (22% [wt/vol]), yeasts *C. deuterogattii* and *C. dubliniensis* (24% and 27% [wt/vol], respectively), and the mold *A. flavus* (>30% [wt/vol]). The susceptibility of microbes to nonperoxide activity was much less variable, ranging from an average of 21% (wt/vol) against the most susceptible (*P. aeruginosa*) to >30% (wt/vol) against the least susceptible (*A. flavus*). In order to assess whether the susceptibility profiles of certain species to honey samples were significantly different from each other, rank correlations were performed (Fig. 1C-D). These were always positive but only reached significance in 12/28 species pairs for total activity and 14/28 for nonperoxide activity, and the correlating pairs were quite different between the different honey activity types. This suggests that while the overall trends in activity remain the same across species, there are species-specific differences in susceptibility.

**Healthy hives produce honey with significantly more phenolics and antioxidant content.** Various chemical properties that have been linked to antimicrobial activity were assayed in the honey samples and compared between healthy and stressed hives (Fig. 2A). Color intensity was generally higher in healthy hive honeys with an average of 441 mAU compared to stressed hive honeys with an average of 386 mAU, but this did not reach significance. Maximum hydrogen peroxide production at 25% honey dilution was not significantly different between healthy and stressed hives and fell within a relatively small range between 49 and 66 with an overall average of 58 $\mu$M. Phenolics content measured by the Folin-Ciocalteu (FC) assay was significantly greater in healthy hive honeys ($P = 0.005$) with an average of 199 and a range of 122 to 252 mg GAE/kg, compared to stressed hive honeys with an average of 138 and a range of 66 to 197 mg GAE/kg. Antioxidant content measured by the ferric reducing antioxidant power (FRAP) assay was also significantly greater in healthy hive honeys ($P = 0.019$) with an average of 3,684 and a range of 2,377 to 4,890 $\mu$mol Fe$^{2+}$/kg, compared to stressed hive honeys with an average of 2,607 and a range of 1,284 to 3,639 $\mu$mol Fe$^{2+}$/kg.

Phenolics and antioxidant content were very strongly positively correlated with each other ($r = 0.938$; $P < 0.001$) suggesting that the majority of antioxidant activity in the honey samples was generated by phenolic compounds (Fig. 2B). Both phenolics content ($r = 0.587$; $P = 0.010$) and antioxidant content ($r = 0.580$; $P = 0.012$) were also positively correlated with color intensity, indicating that phenolic compounds darken the honey. Comparing these properties with antimicrobial activity (Fig. 2C), greater phenolics, and antioxidant content were both significantly correlated with increased nonperoxide activity against *S. aureus* ($P = 0.007$ and $P = 0.005$, respectively) and *E. faecalis* ($P = 0.007$ and $P = 0.009$, respectively), but not against other organisms tested.

**Relative abundances of microbes in the honey bee gut and in hive environment samples.** The bacterial and fungal composition of bee gut samples separated into the crop, midgut and hindgut, and hive environment samples of bee bread, hive entrance swabs, and brood frame swabs, were assessed via 16S and ITS rRNA gene sequencing. In the bacterial gut microbiome (Fig. 3A), the relative abundance of known core bacterial genera averaged across all gut samples was as follows: *Gilliamella* (18%), *Apilactobacillus* (16%), *Lactobacillus* (9%), *Snodgrasella* (8%), *Bombilactobacillus* (5%), *Bombella* (5%), *Bifidobacterium* (4%), *Frischella* (3%), *Commensalibacter* (2%), and *Bartonella* (0.8%). Crop samples were dominated by *Apilactobacillus* (44%) and *Bombella* (16%) and midgut samples by *Gilliamella* (36%) and *Snodgrasella* (15%). Hindgut samples possessed a more even distribution with *Lactobacillus* (15%), *Gilliamella* (14%), *Bombilactobacillus* (12%), *Bifidobacterium* (10%), and *Snodgrasella* (9%) being the most abundant genera. In the fungal gut microbiome, *Starmerella* (7 to 28%) and *Metschinikowia* (9 to 15%) were the

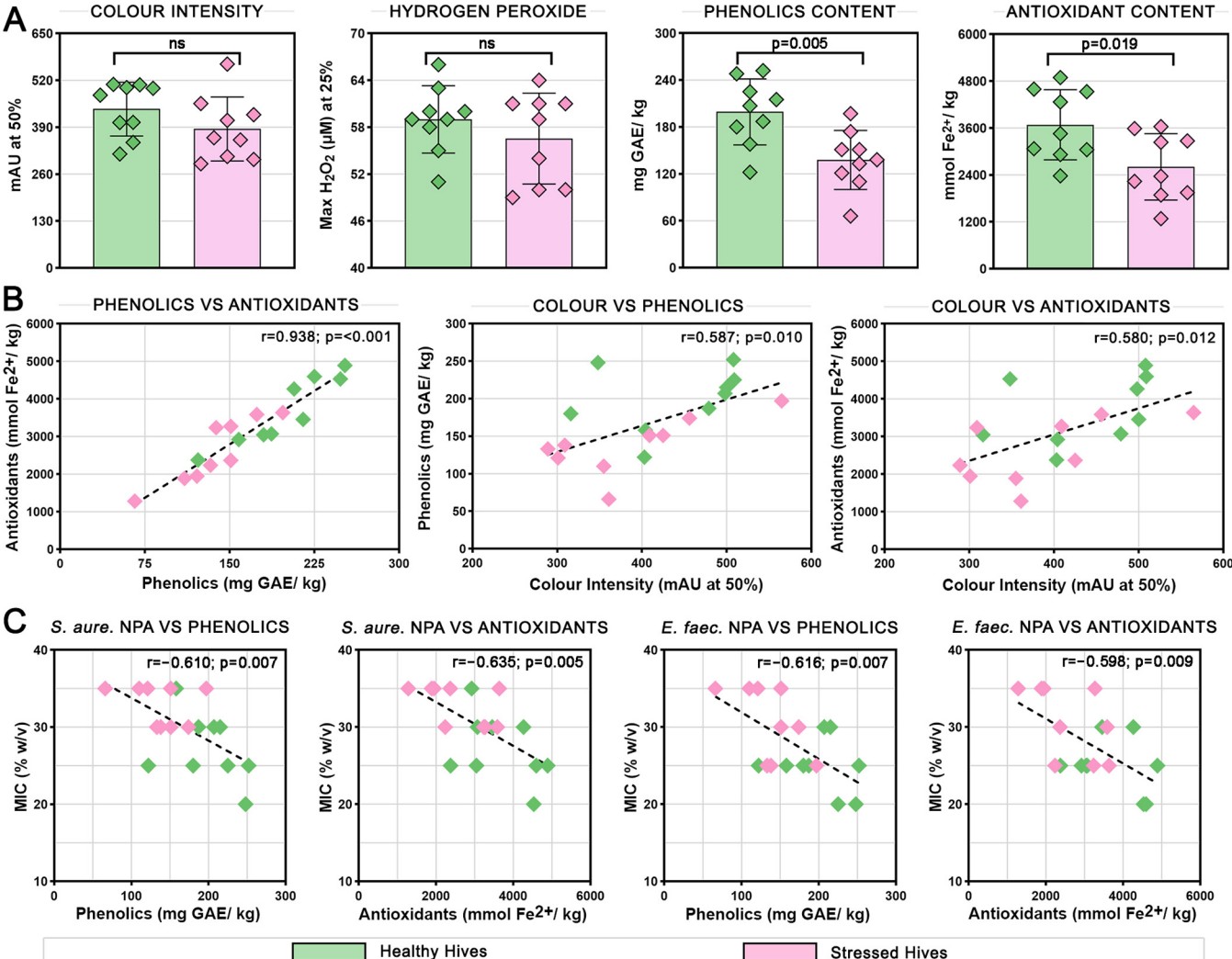

**FIG 2** Chemical properties of honey and their correlation with antimicrobial activity. (A) Color intensity, hydrogen peroxide, phenolics content, and antioxidant content of honey samples from healthy and stressed hives across three time points. Columns represent the mean while diamonds show the values of individual honey samples. Error bars show SEM. (B) Significant pairwise correlations between color intensity, phenolics content, and antioxidant content, and (C) significant correlations between nonperoxide activity (NPA) against *Staphylococcus aureus* and *Enterococcus faecalis* with phenolics and antioxidant content. Lines of best fit, Spearman's rho and *P*-values are shown.

most abundant genera in crop, midgut, and hindgut indicating less distinct differences in fungal community structure between gut sections compared to bacteria.

In the hive samples (Fig. 3B), bee bread samples were dominated by bacterial genera *Rosenbergiella* (22%) and *Acinetobacter* (13%) and by fungal genera *Fusarium* (14%), *Cladosporium* (9%), and *Epicoccum* (9%). Hive entrance and brood frame samples had similar bacteria, with both dominated by *Pseudomonas* (21 to 30%) and *Paenibacillus* (12 to 15%), but were more distinct for fungi with *Meyerozyma* (26%) and *Cladosporium* (14%) being most abundant in hive entrance samples and *Candida* (29%) and *Fusarium* (22%) being most abundant in brood frame samples.

**Alpha diversity values differ with sample type and hive health.** Three measures of alpha diversity were used to compare microbiome samples from healthy and stressed hives: Observed, which is the number of unique ASVs in the sample, the Shannon Index, which gives more weight to species richness, and the Inverse Simpson (InvSimpson) Index, which gives more weight to species evenness. Within the forager gut, gut section was a strong driver of variation in bacterial alpha diversity (Fig. 4A), but not in fungal alpha diversity (Fig. 4B). Observed bacterial alpha diversity was significantly greater in the midgut than the crop (*P* < 0.002) and in the hindgut than the midgut (*P* = 0.012) or crop

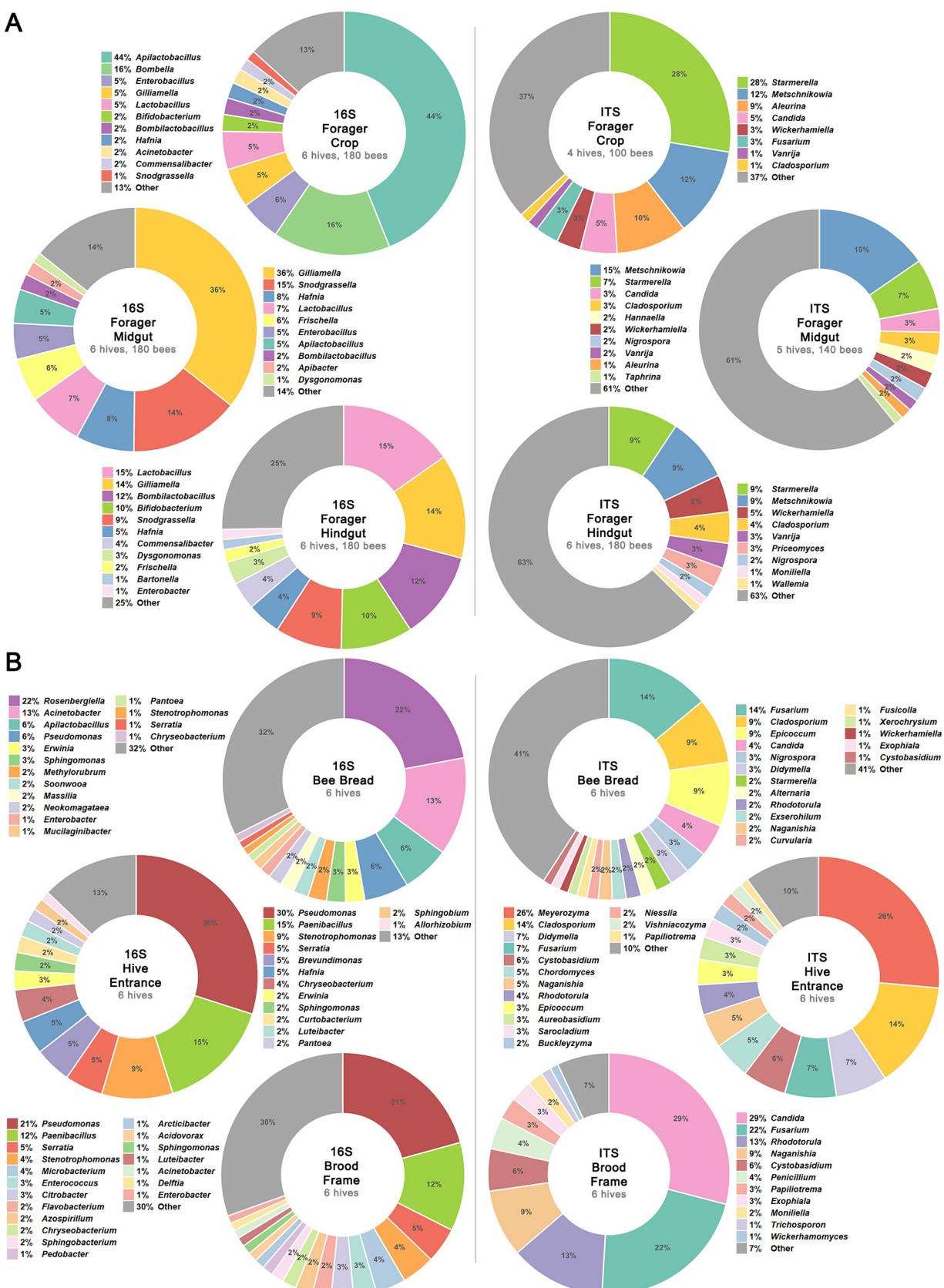

**FIG 3** Relative abundances of microbes in the honey bee gut and hive environment. Relative abundances of bacteria (left) and fungi (right) at genus level in (A) forager bees, including the crop, midgut, and hindgut, and (B) the hive environment, including bee bread stored in the

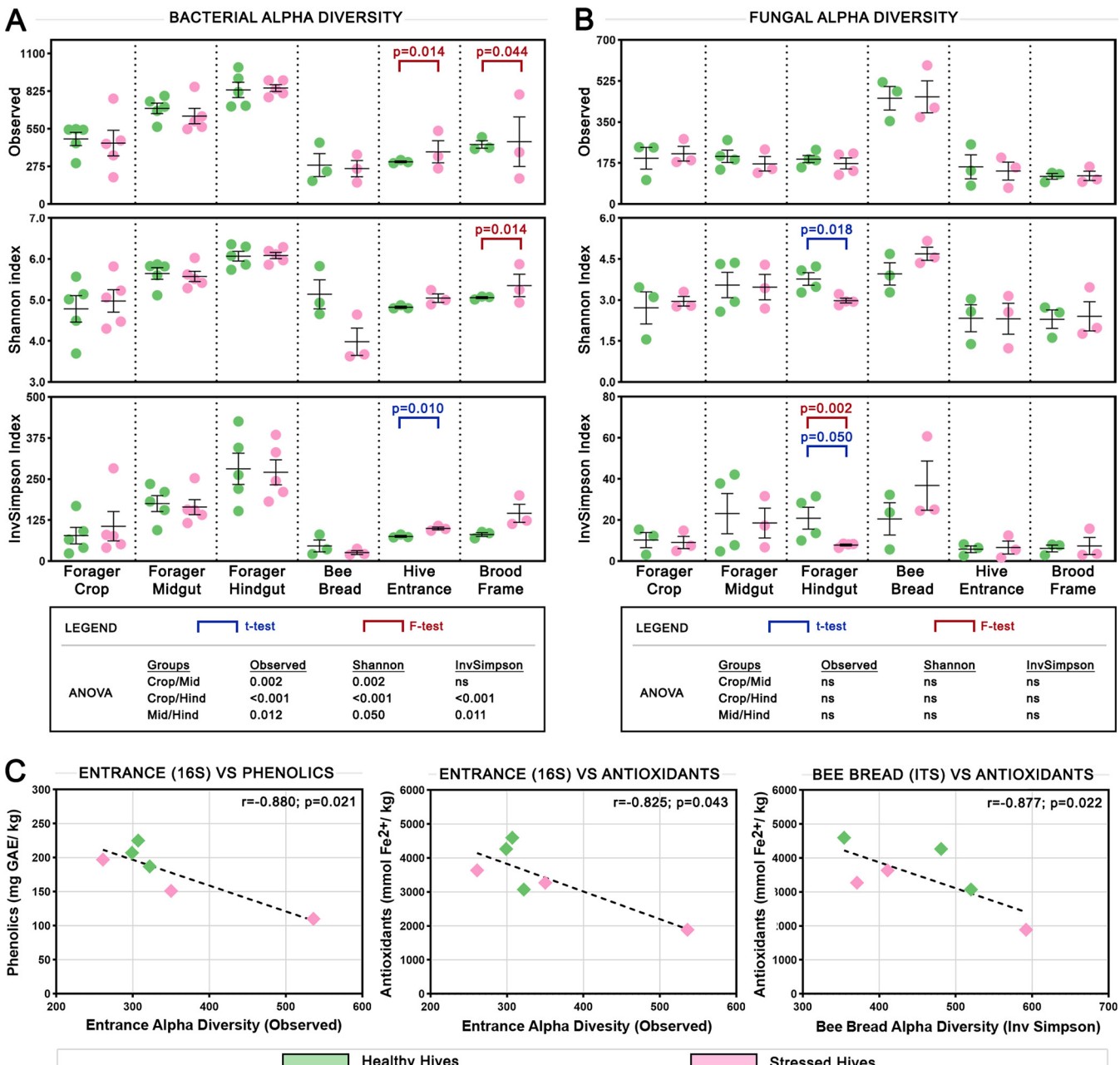

**FIG 4** Alpha diversity values and their correlations with honey properties. (A) Bacterial and (B) fungal alpha diversity at various sites measured using total ASV count (observed), and the Shannon (species richness) and Inverse Simpson's (species evenness) indices. Error bars show the mean and SEM while circles show the values of individual samples. Significant differences between means (*t* test) and variances (F-test) for healthy and stressed samples are shown on the graphs in blue and red, respectively, while *P*-values for comparisons between means (ANOVA) for the three gut sections are shown in the boxes below. (C) Significant correlations between alpha diversity measures with phenolics and antioxidant content; lines of best fit, Spearman's rho and *P*-values are shown.

($P < 0.001$), with the pattern remaining the same for Shannon and InvSimpson alpha diversity. Conversely, there were no significant differences for any measure of fungal alpha diversity between sections of the forager gut.

Comparing bacterial diversity between stressed and healthy hives, stressed hive entrances had significantly more variable observed alpha diversity ($P = 0.014$) and sig-

**FIG 3** Legend (Continued)
hive, the hive entrance, and the brood frames. Values shown are averaged across all samples of one type taken during the study. Taxa with averaged relative abundance ≥1% are shown, while remaining taxa are grouped under 'Other'. Segments without labels have a relative abundance of 1%.

nificantly greater InvSimpson alpha diversity ($P = 0.010$) than healthy hives. Stressed hive brood frames had significantly more variable observed ($P = 0.044$) and Shannon ($P = 0.014$) alpha diversities than healthy hives. These results suggest a link between hive stress and bacterial growth in the hive environment. Looking at fungal diversity, the hindguts of forager bees from healthy hives had significantly greater Shannon alpha diversity ($P = 0.018$) and significantly greater ($P = 0.050$) and more variable ($P = 0.002$) InvSimpson alpha diversity than those from stressed hives. Comparing the alpha diversity of samples to the chemical properties of honey from the same hives revealed significant correlations (Fig. 4C). Phenolics ($r = -0.880$l $P = 0.021$) and antioxidant ($r = -0.825$; $P = 0.043$) content were significantly negatively correlated with bacterial observed alpha diversity at the hive entrance, and antioxidant content was significantly negatively correlated with fungal InvSimpson alpha diversity ($r = -0.877$; $P = 0.022$) in bee bread samples, indicating that phenolics and/or antioxidant content in honey may play a role in preventing microbial overgrowth in the hive environment.

**Beta diversity and relative abundance of core genera differ with hive health.** Multivariate analysis was used to assess differences in microbial profiles between gut segments and hive samples. Within the bacterial microbiome, nonmetric multidimensional scaling (NMDS) based on a Bray-Curtis dissimilarity matrix showed a distinct separation of samples according to sample type, and within the bee samples, according to gut section (Fig. 5A). While gut section samples grouped closely together, and brood frame and hive entrance swab samples grouped together, bee bread samples were more similar to the gut samples than to the swab samples suggesting that the bee bread microbiome is more closely linked to the bees themselves than to the overall hive environment.

ADONIS on the Bray-Curtis dissimilarities showed that overall, 17.4% of variation was significantly accounted for by sample type ($P = 0.001$), and within bee samples, 36.9% of variation was significantly accounted for by gut section ($P = 0.001$). ANOSIM analysis further indicated that there was significantly greater variation between different gut sections than within each section ($R^2=0.710$, $P = 0.001$). Considering hive health alone for ADONIS and ANOSIM yielded no significance, however, a mixed model ADONIS analysis considering both hive health and sample type showed that 6.3% of variation was significantly accounted for ($P = 0.001$). These results indicate that sample type is by far the biggest driver of variation in the bacterial data set, but that after accounting for differences driven by sample type, grouping samples by hive health has significant explanatory power.

Within the fungal microbiome, NMDS showed a much less distinct separation of samples according to sample type (Fig. 5B) compared to the bacterial data set. Additionally, gut section samples were distributed over a much larger area indicating more overall variation in the fungal gut microbiome compared to the bacterial microbiome. ADONIS showed that overall, 21.1% of variation was significantly accounted for considering sample type ($P = 0.001$) and hive health and gut section had no significant explanatory power. Similarly, ANOSIM indicated no significant differences in variation within or between groups according to hive health or gut section, indicating that the fungal communities in each sample type had equal levels of variance.

Comparing relative taxa abundance between samples revealed significant differences between bees from healthy and stressed hives, indicating that differences in community structure aligned with health (Fig. 5C). In the crop, the relative abundance of the bacterial genus *Apilactobacillus* was significantly higher (q = 0.001) in healthy hive bees (average 55%, range 20 to 92%) than in stressed hive bees (average 33%, range 6 to 60%). In the midgut, the relative abundance of bacterial genus *Hafnia* was significantly lower (q = 0.047) in healthy hive bees (average 2%, range 0 to 4%) than in stressed hive bees (average 12%, range 1 to 35%). In the hindgut, the relative abundance of the bacterial genus *Gilliamella* (q = <0.001) was significantly lower in healthy hive bees (average 10%, range 5 to 17%) than in stressed hive bees (average 18%, range 12 to 32%) and the relative abundance of the fungal species *Starmerella lactis-*

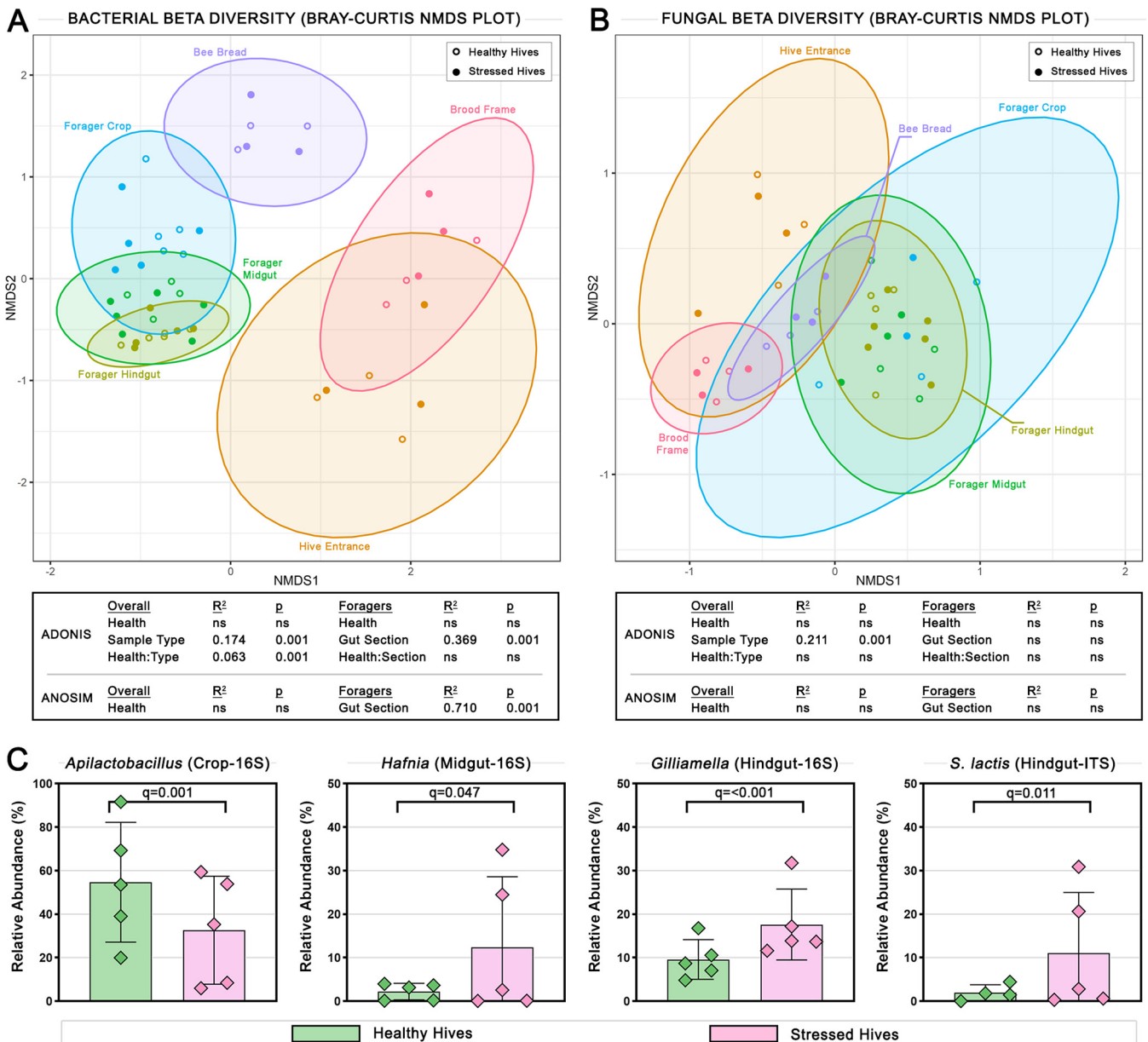

**FIG 5** Beta diversity plots and relative abundance of genera that differ with hive health. (A) Bacterial and (B) fungal beta diversity shown as nonmetric multidimensional scaling (NMDS) plots using Bray-Curtis dissimilarity matrices with the distance between points representing how similar the samples are. Ellipses show the 95% confidence interval of each sample type. $R^2$ and *P*-values for ADONIS analysis showing whether significant amounts of variation are explained by certain groupings, and ANOSIM analysis showing whether samples are significantly more similar within than between groups, are shown in boxes below. Groups under 'Overall' used all samples, including forager gut and hive environment samples, while groups under 'Foragers' used forager gut samples only. (C) Significant differences between the relative abundances of the bacterial genera *Apilactobacillus* in the crop, *Hafnia* in the midgut, and *Gilliamella* in the hindgut, and the fungal species *Starmerella lactis-condensi* in the hindgut in healthy and stressed hives calculated using one-way ANOVAs with Benjamini-Hochberg FDR correction. Columns represent the mean while diamonds show the values of individual samples. Error bars show SEM.

*condensi* (q = 0.011) was significantly lower in healthy hive bees (average 2%, range 0 to 5%) than in stressed hive bees (average 11%, range 0 to 31%).

## DISCUSSION

With increasing threats and sources of stress affecting honey bee populations around Australia, it is more important than ever to understand the full range of impacts that hive stress has on bees and on the hive products that contribute to their health. Many studies compare honey from apiaries in different geographical regions and it is well-established that location, environmental conditions, and floral resources can all play a

significant role in antimicrobial activity (26, 27). Much less is known about the role of entomological differences between hives in honey activity, although substantial differences in activity have been found between honey samples taken from the same apiaries at the same time (26), and knowledge of various bee-derived components that contribute to honey activity suggests that they are important (5).

In the current study, we chose to investigate hive health using hives located near each other in the same apiary, under the assumption that climatic factors would be consistent, and that hives would have access to an identical variety of floral resources. Hives demonstrating signs of relatively minor stress were targeted, and honey and bee samples were taken at three time points over the course of a month with results averaged to minimize the impact of transient outliers in the data. We found that healthy hives produced honey with significantly greater antimicrobial activity against several microbes with average differences in MIC against some species as large as 6%, which is substantial given that the range of MICs for honey is typically between 1 and 30%. Additionally, increased phenolics and antioxidant content in the honey were correlated with hive health, increased nonperoxide antimicrobial activity, and decreased bacterial alpha diversity at the hive entrance and fungal alpha diversity in bee bread. Taken together, these results suggest that healthy hives are producing honey with an increased phenolics content resulting in increased antimicrobial activity and therefore an increased ability to prevent microbial growth in the hive environment, likely feeding back into maintaining hive health.

Phenolics compounds are ubiquitous in the plant kingdom and are acquired by bees via nectar and pollen before incorporation into the honey (28). Given that the same floral resources were available to all hives in this study, stress-induced behaviors such as overall decreased or less selective foraging may explain the lower phenolics content found in honeys from stressed hives. Several previous studies have found various types of hive stress to be linked with altered foraging behavior: foragers who experienced pollen stress during the larval stage displayed reduced foraging activity as adults (29), hives challenged with chalkbrood altered their foraging behavior to increase resin-collection for propolis production (30), and colonies experiencing nutritional stress were found to become less selective and exploit less profitable flower patches (31). Of the three stressed hives in the current study, one was noted to have reduced pollen stores indicating that the hive was undergoing nutritional stress, and all three were noted to have low populations, potentially limiting the foraging capacity of the hive to collect the same variety and quality of floral resources as the healthy hives.

Looking at the hive and bee microbiomes, we found that a significant amount of bacterial variation was explained using a mixed model analysis considering both sample type and hive health, further confirming substantial differences present between groups aligning with stress. Most of the health-related variation appeared to be driven by hive entrance and brood frame samples, which had significantly higher and more variable alpha diversity in stressed hives. This suggests an impaired ability to prevent microbial overgrowth in the hive environment, which could be confirmed by quantitative assessment and culture-based studies of microbial density in the hive environment. In both healthy and stressed hives, *Paenibacillus* and *Serratia* were among the most abundant genera at the hive entrance and in brood frames, and *Hafnia* was also dominant at the hive entrance. The *Paenibacillus* genus includes *Paenibacillus larvae*, the causative agent of the fatal bee disease American Foulbrood, and *Paenibacillus alvei*, associated with European Foulbrood (32). *Serratia* and *Hafnia* are opportunistic pathogens that are also associated with hive or bee disease (33, 34). The apparently large reservoir of these bacteria in the hive environment highlights the critical need for hives to maintain the capacity to prevent the incursion of potentially pathogenic microbes. The significant correlation between decreased alpha diversity in hive environmental samples and increased phenolics and/or antioxidant content in honey samples is intriguing. Whether this is a causal relationship, a by-product of both factors independently aligning with hive health, or an indication that there may be a parallel increase in phenolics content in other hive

products such as propolis that function more directly to keep pathogens out, is unknown and worthy of further study in future analyses.

Significant differences were found in the relative abundance of core and opportunistically pathogenic taxa present in the bee gut. While the crops of bees from both healthy and stressed hives were dominated by *Apilactobacillus*, significantly more was present in healthy bee crops. The predominance of *Apilactobacillus*, a genus recently reclassified from *Lactobacillus* (35), is consistent with other studies looking at the crop (36) where the main species of *Apilactobacillus* is the host-adapted *A. kunkeii*. With the microbial composition of the crop considered to be more transient and reflective of exposure to nectar and the environment, and the fact that *Apilactobacillus* also contains many species found in the sugar-rich environments of flowers and fruits, these results could be further evidence of differential foraging behavior between bee colonies in healthy and stressed hives. *Apilactobacillus* spp within the gut facilitate carbohydrate metabolism and produce important secondary metabolites (37), and their increased abundance in healthy bee crops where nectar and honey are stored could be a potential source of additional antimicrobial or other health-promoting compounds in healthy hive honey.

In the midgut, significantly more *Hafnia-Obsesumbacterium* was present in stressed hives. Members of this genus are known to act as opportunistic pathogens in hives experiencing stress, with the species *Hafnia alvei* implicated in septicemia in bees, causing high mortality (33). In the hindgut, significantly more *Gilliamella* was present in bees from stressed hives. *Gilliamella* is one of the core phylotypes present in all adult honey bee guts (20) and is implicated in the degradation of dietary polysaccharides (38), mediation of immune defense pathways (39), and differential gene expression related to social responsiveness (40). A study looking at hives with high and low honey productivity found that less productive hives were more likely to be dominated by *Gilliamella* while more productive hives had a higher abundance of *Lactobacillus* (41). Another study using gnotobiotic bees found that those colonized with *Gilliamella* exhibited decreased expression of major royal jelly protein genes (40). The major royal jelly proteins have polyfunctional properties, including being the precursor of antimicrobial peptides (14), and a potential decrease in their expression could be related to the decreased nonperoxide activity seen in honey from stressed hives. Thus, a higher relative abundance of *Gilliamella* in the hindgut of bees from stressed hives may be an indicator of a less favorable gut microbial community structure, with potential links to the properties of honey produced.

The fungal microbiome of the gut appeared less structured and with substantial variation in species composition compared to the bacterial microbiome, with gut section unable to explain any significant variation between samples. Other studies have found conflicting results: one reported a similarly high level of variability in the composition of the fungal microbiome suggesting that fungi might be transient rather than core (42), while another reported a clear gradient between gut compartments indicating a more stable community structure (43). Nonetheless, gut fungi are thought to perform unique functions beneficial to the bee host, including sugar fermentation, nutrient recycling, and long-term preservation of pollen (44).

In the crop, midgut, and hindgut we found *Starmerella* and *Metschnikowia* to be the predominant fungal genera with a significantly increased presence of *Starmerella lactiscondensi* in the hindgut of bees from stressed hives. *Starmerella* is a genus of sugar tolerant, fermentative yeasts and is frequently associated with bee bread (45), but variably present in the gut. Recent studies found it among the most abundant genera in Italian (43) and Polish (46) bees, but not in Californian (42) or Virginian (47) bees. Brazilian stingless bees appear to be associated with yeast communities composed predominantly of *Starmerella* (48), and in a study of bees experiencing an outbreak of disease, a decrease in the relative abundance of *Starmerella* and other dominant fungi was associated with pronounced shifts in diet, including a switch from native plant to *Eucalyptus* pollen, an exotic tree in Brazil (49). *Eucalyptus* is native to Australia and is predominant in the area surrounding the hives sampled in this study, thus the significant difference observed for

**TABLE 1** Characteristics of the six hives used in this study

| Hive no. | Status | Beekeeper assessment |
|---|---|---|
| Hive 1 | Healthy | High honeybee population. Queen present with good brood pattern. No signs of disease. |
| Hive 2 | Healthy | High honeybee population. Queen present with good brood pattern. No signs of disease. |
| Hive 5 | Healthy | High honeybee population. Queen present with good brood pattern. No signs of disease. |
| Hive 3 | Stressed | Low honeybee population. Queenless with poor brood pattern. No signs of disease. |
| Hive 9 | Stressed | Low honeybee population. Queen present with good brood pattern. Signs of chalkbrood and pest larvae. |
| Hive CB1 | Stressed | Low honeybee population. Queen present with poor brood pattern and new queen cells developing. Reduced pollen stores. |

*Starmerella lactis-condensi* in bees from healthy and stressed hives could reflect differences in *Eucalyptus* foraging behavior.

**Conclusions.** This study is the first to show a direct connection between hive health and the antimicrobial activity of honey. Our results underscore the need for understanding and proactive management of honey bee health by beekeepers, as we found even apparently minor stress can have implications for overall hive fitness as well as the economic potential of hive products. A limitation of our work is that it is retrospective and across a relatively short timescale, and we can only speculate on the factors that may have caused hive decline. Future studies using hives undergoing controlled experimental treatments that could cause measurable differences to honey bee health will be instrumental in building upon these foundational results.

## MATERIALS AND METHODS

**Sample collection.** Samples were collected from an apiary at Tocal Agricultural College Bee Research and Training Centre in Paterson, New South Wales (NSW). Beekeeper assessment was used to select hives, considering factors, including population size, queen presence, brood pattern, and signs of disease. The three chosen healthy hives had high populations, queens present, good brood patterns, and no signs of disease, while factors affecting stressed hives included low bee populations, absent queens, poor brood patterns, and signs of chalkbrood (Table 1). Sampling was performed on three occasions at 2-week intervals on 17 March 2022, 31 March 2022, and 14 April 2022. From each hive, forager bees were collected into 100% ethanol from around the entrance, bee bread was collected from cells in the brood frames, and mature, capped honey was collected from the center of honey frames. The hive entrance and brood frames were each swabbed twice with sterile swabs and stored in Amies transport medium. After transport, forager bees, bee bread, and swabs were stored at −80°C until use, while honey samples were stored in the dark at 4°C.

**Antimicrobial susceptibility testing.** Antimicrobial susceptibility testing by broth microdilution was performed in accordance with CLSI guidelines with results expressed as minimal inhibitory concentration (MIC). Honeys were assayed at 5, 10, 15, 20, 25, and 30% (wt/vol) diluted in either sterile water for total activity or freshly prepared 5600 U/mL catalase solution for nonperoxide activity. Full details of test strains, culture conditions, and incubation conditions are outlined in Supplemental Material S1.

**Assessment of honey color.** The absorbance of honey samples at 50% (wt/vol) was measured at 450 and 720 nm using a UV/Vis spectrophotometer (Specord S600) in quartz cuvettes with 10 mm optical pathlength. Color intensity was calculated using the equation $Colour\ Intensity\ (mAU) = (A_{720} - A_{450}) \times 1000$.

**Amplex Red hydrogen peroxide assay.** The Amplex Red (Thermo Fisher A22188) hydrogen peroxide assay was performed according to the manufacturer's instructions. Honeys were assayed at 25% (wt/vol) and measurements were taken at 535 nm every 30 min for 24 h. $H_2O_2$ standards ranging from 0.5 – 1024 $\mu$M were used to generate a standard curve and the resulting equation from the line of best fit was used to calculate the amount of $H_2O_2$ in each honey sample.

**Folin-Ciocalteu (FC) total phenolics assay.** In a 96-well plate, 20 $\mu$L aliquots of 20% (wt/vol) honey samples were prepared in triplicate. To each sample, 100 $\mu$L of FC reagent (1 mL Folin-Ciocalteu reagent in 30 mL sterile water) was added with incubation in the dark for 5 min at RT, followed by 80 $\mu$L of 0.75% $Na_2CO_3$ with incubation in the dark for 90 min at RT. Absorbance was measured at 760 nm using a plate reader. Gallic acid standards ranging from 0.06 to 0.18 mg/mL were used to generate a standard curve and the resulting equation for the line of best fit was used to calculate the phenolics content of honey samples, expressed as mg of gallic acid equivalent per kg of honey (mg GAE/kg).

**Ferric reducing antioxidant power (FRAP) assay.** FRAP reagent (a 1:1:10 (vol/vol/vol) ratio of 10 mM 2,4,6-tris(2-pyridyl)-s-triazine dissolved in 40 mM HCl, 20 mM $FeCl_3$, and 300 mM pH 3.6 acetate buffer) was prepared fresh and warmed to 37°C prior to the assay. In a 96-well plate, 20 $\mu$L aliquots of 20% (wt/vol) honey samples were prepared in triplicate. To each sample, 180 $\mu$L of FRAP reagent was added with incubation for 30 min at 37°C. Absorbance was measured at 594 nm using a plate reader. $FeSO_4$ standards ranging from 200 – 1200 $\mu$M, made fresh and stored on ice until use, were used to generate a standard curve and the resulting equation from the line of best fit was used to calculate FRAP value, expressed as $\mu$mol $Fe^{2+}$/kg.

**DNA preparation and amplicon sequencing.** Honey bee guts were extracted and separated into the crop, midgut, and hindgut. Gut dissections were performed for 20 bees per hive and pooled. For bee bread samples, 100 mg was used, and for swab samples, swab heads were cut off using sterile scissors,

removing any hard plastic pieces. Samples were homogenized in a bead beater and DNA extractions were performed using DNeasy Blood & Tissue kits (Qiagen) according to the manufacturer's instructions. DNA was sent to Ramaciotti Centre for Genomics at the University of New South Wales, Sydney for 16S rRNA V3-V4 amplicon sequencing with the 341F-805R primer set using the Illumina Miseq v3 $2 \times 300$ bp platform, and to BGI Genomics, Hong Kong for ITS1 amplicon sequencing with the ITS1F-ITS2 primer set using the DNBSEQ PE300 platform. Final sample numbers were as follows: $n = 6$ for hive environment samples, $n = 10$ for 16S bee gut samples, and $n = 6$ to 8 for ITS bee gut samples. Raw sequence reads were processed in R v4.2.2 using the DADA2 pipeline to generate amplicon sequence variants (ASVs). Taxonomy was assigned using the SILVA database release 138.1 for 16S and the UNITE database release 27.10.2022 for ITS. Full details of sample preparation, DNA extraction, and data processing are outlined in Supplemental Material S1.

**Statistical analysis.** The Shapiro-Wilk normality test was used to determine whether the data were normally distributed. Significant differences between groups for parametric data were determined using t-tests for 2 groups, or an ANOVA with *post hoc* Tukey-Kramer test for >2 groups. Significant differences between groups for nonparametric data were determined using Mann-Whitney U tests for 2 groups, or Kruskal-Wallis H tests with *post hoc* Dunn's tests for >2 groups. Differences in variance were assessed by F-tests. Associations between parametric variables were assessed using Pearson's product-moment correlations, and associations between nonparametric or parametric and nonparametric variables were assessed using Spearman's rank correlations. Amplicon data (ASVs) were rarefied to even sampling depth prior to the calculation of alpha diversity metrics and were subject to centered-log ratio transformation for beta diversity measures. For alpha diversity, total ASV count (observed) and the Shannon and Inverted Simpson indices were calculated using the phyloseq R package. For beta diversity, Bray-Curtis dissimilarity matrices were calculated and nonmetric multidimensional scaling (NMDS) was performed using the phyloseq R package. Permutational ANOVAs (ADONIS) and analysis of similarities (ANOSIM) were performed using the R vegan package. For comparisons of relative abundance, one-way ANOVAs with Benjamini-Hochberg FDR correction were used. Unless otherwise specified, two-tailed $P$ values were used for all tests, and $P$ values <0.05 were considered significant. Error bars represent the mean +/-95% standard error of measurement (SEM). Data were analyzed and visualized using Excel (Microsoft Corporation), Prism 9 (GraphPad Inc.), and R v4.2.2 software.

**Data accessibility statement.** Raw metagenomic data obtained during this study is publicly available in the NCBI Sequence Read Archive under Bioproject ID PRJNA975210.

## SUPPLEMENTAL MATERIAL

Supplemental material is available online only.
**SUPPLEMENTAL FILE 1**, DOCX file, 0.03 MB.
**SUPPLEMENTAL FILE 2**, XLSX file, 0.01 MB.

## ACKNOWLEDGMENTS

Honey research projects undertaken by our team are supported by the NSW Bushfire Industry Recovery Package Sector Development Grant (BIP-SDG-135). We thank Stanislav Nenov (NSW Department of Primary Industries) for his assistance with sample collection.

K.E.F. and D.A.C. conceived and designed the experiments. K.E.F., B.S., D.S., and D.A.C. collected the samples with assistance and hive health assessment from E.A.F.; K.E.F. and B.S. produced the honey activity and chemistry data with assistance from D.S., A.Z.D., and T.D.T.; K.E.F. and B.S. produced the microbiome data with technical assistance from A.Z.D. and bioinformatics assistance from E.R.S.; K.E.F. collated and analyzed the data and wrote the manuscript with assistance from B.S., D.A.C., and N.N.C.

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
