## [Reviewer comments · Microbiology Spectrum]

Microbiology Spectrum

Low levels of hive stress are associated with decreased honey activity and changes to the gut microbiome of resident honey bees

Kenya Fernandes, Bridie Stanfield, Elizabeth Frost, Erin Shanahan, Daniel Susantio, Andrew Dong, Trong Tran, Nural Cokcetin, and Dee Carter

Corresponding Author(s): Dee Carter, The University of Sydney

Review Timeline:

Submission Date:	February 21, 2023
Editorial Decision:	May 1, 2023
Revision Received:	May 10, 2023
Accepted:	May 18, 2023

Editor: Jeffrey Gralnick

Reviewer(s): Disclosure of reviewer identity is with reference to reviewer comments included in decision letter(s). The following individuals involved in review of your submission have agreed to reveal their identity: Duan C Copeland (Reviewer #2)

Transaction Report:

DOI: <https://doi.org/10.1128/spectrum.00742-23>

April 13, 2023

Prof. Dee A Carter
The University of Sydney
School of Life And Environmental Sciences
LEES Building F22, School of Life And Environmental Sciences
University of Sydney
Sydney, New South Wales 2006
Australia

Re: Spectrum00742-23 (Low levels of hive stress are associated with decreased honey activity and changes to the gut microbiome of resident honey bees)

Dear Prof. Dee A Carter:

Thank you for submitting your manuscript to Microbiology Spectrum. While Reviewer 1 identified some grammatical issues with the text, Reviewer 2 suggests that the microbiome data be re-analyzed with a more appropriate workflow and database (BEEexact). While this would indeed be a significant amount of extra work, it may ultimately result in a better representation of the microbiome in your samples. If you have a strong argument against using the newer database, I would certainly consider it.

Link Not Available

Sincerely,

Jeffrey Gralnick

Senior Editor, Microbiology Spectrum

Journals Department
Reviewer comments:

Reviewer #1 (Comments for the Author):

Reviewer Remarks to the Author:

I read with interest the paper titled "Low levels of hive stress are associated with decreased honey activity and changes to the

gut microbiome of resident honey bees." Briefly, the paper linked the hive health to antimicrobial activity of honey.

However, there are small typos that can be edited.

Specific comments:

1. Line 76-77 need to be edited to bring out the meaning of the sentence "....., and the ecological and economic imperative to understand and protect bees from hive stress,"
2. Line 99 an apostrophe needs to be inserted in honeys' "Healthy hive honeys in general had lower MICs...."
3. Line 129-130: The authors could explain the discrepancies in the results on phenolics, i.e., what could be the reason for variation in phenolics in HPLC (figure S3) and spectrophotometric method (figure 2A).
4. Line 623: The legend of supplementary figure S3 need to be expound considering the colour and abbreviations just like the other legends used in the manuscript.
5. For the purposes of review, could the author also provide the supplementary figures Fig S1 and 2. Or Fig S3. Was erroneously labelled?

Reviewer #2 (Comments for the Author):

REVIEW - Low levels of hive stress are associated with decreased honey activity and changes to the gut microbiome of resident honey bees

Fernandes et.al investigated the antimicrobial activity, chemical properties, and microbiome of honey bees and their hives, comparing healthy hives to those showing signs of stress. Honey from healthy hives exhibited higher antimicrobial activity, which was linked to increased phenolics and antioxidant content. The microbiome portion suffers from methods which may obscure significant results (pooling bees, genus-level classification w/Silva database). Some of the reported results are contrary to well established bee gut microbiome studies. Overall, the manuscript was very well written, easy to read and understand, but the microbiome portion would need to be re-analyzed resulting in many of the statistics and figures needing to be recreated.

Major Issues:

- I very much appreciated the authors, splitting tissues into crops, midguts, and hindgut, however I'd argue pooling is good for screening samples (toxin effects, virus titers), but less so for subtle differences in the microbiome. Because the design was such that foragers were randomly taken without accounting for age it is difficult to consider how real the differences are between healthy and stressed colonies. Like other animal systems, the honey bee gut microbiome ages as well.
- Honey bee microbiome research has advanced far enough that we can more accurately assign ASV to the species-level. Simply, genus-level classification is not appropriate at this stage. See BEEexact for accurately classifying honey bee bacteria to species-level, which substantially improves classification over Silva database: DOI:10.1128/msystems.00082-21
- Did the authors control for contamination in extraction and sequencing? Red flags for me were 25% Other in the Hindgut. The proportion of "Other" bacteria in each sequenced niche is inexplicable when studies cited here report that the "core" microbiome usually makes up >90% of reads. I have not seen *Dysgonomonas* in honey bees at the levels reported here. The % relative abundance of *Hafnia* being higher than other core microbial members. For a way to remove contams from dataset See decontam: <https://github.com/benjjneb/decontam> but this also might be artifacts from using Silva over BEEexact

Minor Issues:

- It's important to have the databases you use in the main manuscript instead of supplement.
- Line 269: Likely *Paenibacillus alvei*, unless your hives were experiencing an AFB outbreak.
- Line 285: The main *Apilactobacillus* in honey bees is *A. kunkeei* which is incredibly host-adapted. These are likely not the same one's from the foraging environment.
- Line 386 "For bee bread samples, 100 mg was used and for swab samples, swab heads were cut off using sterile scissors, removing any hard plastic pieces.", add a comma after "used" for improved readability
- Line 393 "Final sample numbers were n=6 for hive environment samples, n=10 for 16S bee gut samples, and n=6-8 for ITS bee gut samples." to "Final sample numbers were as follows: n=6 for hive environment samples, n=10 for 16S bee gut samples, and n=6-8 for ITS bee gut samples." This improves clarity and readability.
- Line 399: Replace "Whether data were normally distributed was determined using the Shapiro-Wilk normality test." with "The Shapiro-Wilk normality test was used to determine whether the data were normally distributed."
- In the sentence "Amplicon data (ASVs) were rarefied to even sampling depth prior to the calculation of alpha diversity metrics, or subject to centred-log ratio transformation for beta diversity measures.", change "or" to "and" to clarify that both methods were used in the analysis.
- Line 403 Kruskal-Wallis should be Kruskal-Wallis, in general fix small misspellings throughout manuscript.

Staff Comments:

Preparing Revision Guidelines

Please return the manuscript within 60 days; if you cannot complete the modification within this time period, please contact me. If you do not wish to modify the manuscript and prefer to submit it to another journal, please notify me of your decision immediately so that the manuscript may be formally withdrawn from consideration by Microbiology Spectrum.

1 Low levels of hive stress are associated with decreased honey 2 activity and changes to the gut microbiome of resident honey bees

Kenya E. Fernandes¹, Bridie Stanfield¹, Elizabeth A. Frost^{2,3}, Erin R. Shanahan^{1,4}, Daniel Susantio¹,
Andrew Z. Dong¹, Trong D. Tran⁵, Nural N. Cokcetin⁶ & Dee A. Carter^{1,7#}

¹School of Life and Environmental Sciences, University of Sydney, Sydney, NSW, Australia

²ABGU, A Joint Venture of NSW Department of Primary Industries and University of New England, Armidale, NSW,
Australia

³NSW Department of Primary Industries, Paterson, NSW, Australia

⁴Charles Perkins Centre, University of Sydney, Sydney, NSW, Australia

⁵School of Science, Technology and Engineering, University of the Sunshine Coast, Maroochydore, QLD, Australia

⁶Australian Institute for Microbiology and Infection, University of Technology, Sydney, NSW, Australia

⁷Sydney Institute for Infectious Diseases, University of Sydney, Sydney, NSW, Australia

**Running Head:** Hive stress, honey activity, and the bee gut

**#Correspondence:** Dee A. Carter, dee.carter@sydney.edu.au

**No. of Words in Abstract:** 164

**No. of Words in Importance:** 146

**No. of Words in Main Text:** 4999

20 **ABSTRACT**

[revised manuscript text omitted]

The antimicrobial activity of honey samples from each hive across the three timepoints were tested
against a range of pathogenic microbes including bacteria, yeasts, and moulds in order to determine
whether there were differences in activity and if these aligned with hive health. Both total activity, a
test of the honey's overall activity including hydrogen peroxide-based action, and non-peroxide
activity, a test of the honey's remaining activity after catalase has been used to abolish the action of
hydrogen peroxide, were determined. Individual values for each honey sample as well as artificial
and control honeys are shown in Supplementary Table S2. Fig. 1A-B compares the average activity
of honey samples from healthy vs stressed hives. Healthy hive honeys in general had lower MICs
than stressed hive honeys and were significantly more active against *E. faecalis* (3% (w/v) lower;
$p=0.002$) and *C. deuterogattii* (5% (w/v) lower; $p=0.026$) for total activity, and against *S. aureus*
(6% (w/v) lower; $p=0.005$), *E. faecalis* (6% (w/v) lower; $p=0.001$), and *P. aeruginosa* (3% (w/v)
lower; $p=0.036$) for non-peroxide activity.

The MIC for artificial honey, a sugar solution used as a control for the osmolarity of honey, was
>35% (w/v) for all species tested except the Gram-negative bacterium *P. aeruginosa*, which
displayed increased susceptibility to sugar with an MIC of 25% (w/v) (Table S1). *P. aeruginosa*
was also the most susceptible microbe to total honey activity (on average 12% (w/v)), followed by
Gram-positive bacteria *S. aureus* and *E. faecalis* and the fungal dermatophyte *T. interdigitale* (18%
(w/v)), Gram-negative bacterium *E. coli* (22% (w/v)), yeasts *C. deuterogattii* and *C. dublinensis*
(24% and 27% (w/v), respectively), and the mould *A. flavus* (32% (w/v)). The susceptibility of
microbes to non-peroxide activity was much less variable, ranging from an average of 21% (w/v)
against the most susceptible (*P. aeruginosa*) to 33% (w/v) against the least susceptible (*A. flavus*).
In order to assess whether the susceptibility profiles of certain species to honey samples were
significantly different from each other, rank correlations were performed (Fig. 1C-D). These were
always positive but only reached significance in 12/28 species pairs for total activity and 14/28 for
non-peroxide activity, and the correlating pairs were quite different between the different honey
activity types. This suggests that while the overall trends in activity remain the same across species,
there are species-specific differences in susceptibility to certain samples.

**Healthy hives produce honey with significantly more phenolics and antioxidant content**

Various chemical properties that have been linked to antimicrobial activity were assayed in the
honey samples and compared between healthy and stressed hives (Fig. 2A). Colour intensity was
generally higher in healthy hive honeys with an average of 441 mAU compared to stressed hive
honeys with an average of 386 mAU, but this did not reach significance. Maximum hydrogen
peroxide production at 25% honey dilution was not significantly different between healthy and
stressed hives and fell within a relatively small range between 49 – 66 with an overall average of 58
μM . Phenolics content measured by the Folin-Ciocalteu (FC) assay was significantly greater in
healthy hive honeys ($p=0.005$) with an average of 199 and a range of 122 – 252 mg GAE/kg,
compared to stressed hive honeys with an average of 138 and a range of 66 – 197 mg GAE/kg.
Antioxidant content measured by the ferric reducing antioxidant power (FRAP) assay was also
significantly greater in healthy hive honeys ($p=0.019$) with an average of 3684 and a range of 2377
128 – 4890 $\mu\text{mol Fe}^{2+}/\text{kg}$, compared to stressed hive honeys with an average of 2607 and a range of
129 1284 – 3639 $\mu\text{mol Fe}^{2+}/\text{kg}$. HPLC analysis of honey samples revealed no apparent differences in
phenolics profile (Supplementary Figure S3).

[revised manuscript text omitted]

**Statistical Analysis**

Whether data were normally distributed was determined using the Shapiro-Wilk normality test.
Significant differences between groups for parametric data were determined using t-tests for 2

groups, or an ANOVA with post-hoc Tukey-Kramer test for >2 groups. Significant differences
between groups for nonparametric data were determined using Mann-Whitney U tests for 2 groups,
or Kruskal-Wallis H tests with post-hoc Dunn's tests for >2 groups. Differences in variance were
assessed by F-tests. Associations between parametric variables were assessed using Pearson's
product-moment correlations, and associations between nonparametric or parametric and
nonparametric variables were assessed using Spearman's rank correlations. Amplicon data (ASVs)
were rarefied to even sampling depth prior to the calculation of alpha diversity metrics, or subject to
centred-log ratio transformation for beta diversity measures. For alpha diversity, total ASV count
(observed) and the Shannon and Inverted Simpson indices were calculated using the phyloseq R
package. For beta diversity, Bray-Curtis dissimilarity matrices were calculated and non-metric
multidimensional scaling (NMDS) was performed using the phyloseq R package. Permutational
ANOVAs (ADONIS) and analysis of similarities (ANOSIM) were performed using the R vegan
package. For comparisons of relative abundance, one-way ANOVAs with Benjamini-Hochberg
FDR correction were used. Unless otherwise specified, two-tailed p values were used for all tests,
and p values <0.05 were considered significant. Error bars represent the mean +/- 95% standard
error of measurement (SEM). Data were analysed and visualised using Excel (Microsoft
Corporation), Prism 9 (GraphPad Inc), and R v4.2.2 software.

**ACKNOWLEDGEMENTS**

Honey research projects undertaken by our team are supported by the NSW Bushfire Industry
Recovery Package Sector Development Grant (BIP-SDG-135). We thank Stanislav Nenov (NSW
Department of Primary Industries) for his assistance with sample collection.

**AUTHOR CONTRIBUTIONS**

KEF and DAC conceived and designed the experiments. KEF, BS, DS, and DAC collected the
samples with assistance and hive health assessment from EAF. KEF and BS produced the honey
activity and chemistry data with assistance from DS, AD, and TDT. KEF and BS produced the
microbiome data with technical assistance from AD and bioinformatics assistance from ERS. KEF
collated and analysed the data and wrote the manuscript with assistance from BS, DAC and NNC.

**DATA ACCESSIBILITY STATEMENT**

Upon acceptance of the manuscript, amplicon sequences will be made available on the NCBI
sequence read archive and metadata related to the publication made available on Dryad.

[revised manuscript text omitted]

Fernandes et.al investigated the antimicrobial activity, chemical properties, and microbiome of honey bees and their hives, comparing healthy hives to those showing signs of stress. Honey from healthy hives exhibited higher antimicrobial activity, which was linked to increased phenolics and antioxidant content. The microbiome portion suffers from methods which may obscure significant results (pooling bees, genus-level classification w/Silva database). Some of the reported results are contrary to well established bee gut microbiome studies. Overall, the manuscript was very well written, easy to read and understand, but the microbiome portion would need to be re-analyzed resulting in many of the statistics and figures needing to be recreated.

Major Issues:

- I very much appreciated the authors, splitting tissues into crops, midguts, and hindgut, however I'd argue pooling is good for screening samples (toxin effects, virus titers), but less so for subtle differences in the microbiome. Because the design was such that foragers were randomly taken without accounting for age it is difficult to consider how real the differences are between healthy and stressed colonies. Like other animal systems, the honey bee gut microbiome ages as well.
- Honey bee microbiome research has advanced far enough that we can more accurately assign ASV to the species-level. Simply, genus-level classification is not appropriate at this stage. See BEEexact for accurately classifying honey bee bacteria to species-level, which substantially improves classification over Silva database:
DOI:10.1128/msystems.00082-21
- Did the authors control for contamination in extraction and sequencing? Red flags for me were 25% Other in the Hindgut. The proportion of "Other" bacteria in each sequenced niche is inexplicable when studies cited here report that the "core" microbiome usually

makes up >90% of reads. I have not seen *Dysgonomonas* in honey bees at the levels reported here. The % relative abundance of *Hafnia* being higher than other core microbial members. For a way to remove contams from dataset See decontam: <https://github.com/benjineb/decontam> but this also might be artifacts from using Silva over BEEExact

•

Minor Issues:

- It's important to have the databases you use in the main manuscript instead of supplement.
- Line 269: Likely *Paenibacillus alvei*, unless your hives were experiencing an AFB outbreak.
- Line 285: The main *Apilactobacillus* in honey bees is *A. kunkeei* which is incredibly host-adapted. These are likely not the same one's from the foraging environment.
- Line 386 "For bee bread samples, 100 mg was used and for swab samples, swab heads were cut off using sterile scissors, removing any hard plastic pieces.", add a comma after "used" for improved readability
- Line 393 "Final sample numbers were n=6 for hive environment samples, n=10 for 16S bee gut samples, and n=6-8 for ITS bee gut samples." to "Final sample numbers were as follows: n=6 for hive environment samples, n=10 for 16S bee gut samples, and n=6-8 for ITS bee gut samples." This improves clarity and readability.
- Line 399: Replace "Whether data were normally distributed was determined using the Shapiro-Wilk normality test." with "The Shapiro-Wilk normality test was used to determine whether the data were normally distributed."
- In the sentence "Amplicon data (ASVs) were rarefied to even sampling depth prior to the calculation of alpha diversity metrics, or subject to centred-log ratio transformation for beta diversity measures.", change "or" to "and" to clarify that both methods were used in the analysis.
- Line 403 Kruskall-Wallis should be Kruskal-Wallis , in general fix small misspellings throughout manuscript.

We thank both reviewers for their helpful comments and suggestions which we have taken on-board to improve the manuscript. We address each of these below.

Reviewer #1

I read with interest the paper titled "Low levels of hive stress are associated with decreased honey activity and changes to the gut microbiome of resident honey bees." Briefly, the paper linked the hive health to antimicrobial activity of honey.

However, there are small typos that can be edited.

Specific comments:

1. Line 76-77 need to be edited to bring out the meaning of the sentence "....., and the ecological and economic imperative to understand and protect bees from hive stress,"

Changed to "as well as the ecological and economic imperatives to understand and protect bees from hive stress" which we believe flows better.

2. Line 99 an apostrophe needs to be inserted in honeys' "Healthy hive honeys in general had lower MICs...."

Changed to "honey from healthy hives in general had lower MICs than honey from stressed hives".

3. Line 129-130: The authors could explain the discrepancies in the results on phenolics, i.e., what could be the reason for variation in phenolics in HPLC (figure S3) and spectrophotometric method (figure 2A).

On further consideration we have decided to remove the supplementary HPLC data from the manuscript as we are not sure of its accuracy, and we were unable to pursue further in-depth follow-up analysis due to some honey samples running out.

4. Line 623: The legend of supplementary figure S3 need to be expound considering the colour and abbreviations just like the other legends used in the manuscript.

We have decided to remove the supplementary HPLC data from the manuscript.

5. For the purposes of review, could the author also provide the supplementary figures Fig S1 and 2. Or Fig S3. Was erroneously labelled?

The manuscript now includes Supplementary Material S1 (supplementary methods document), and Supplementary Table S2 (spreadsheet of all data). There are no Supplementary Figures S1 and S2.

Reviewer #2

Fernandes et. al investigated the antimicrobial activity, chemical properties, and microbiome of honey bees and their hives, comparing healthy hives to those showing signs of stress. Honey from healthy hives exhibited higher antimicrobial activity, which was linked to increased phenolics and antioxidant content. The microbiome portion suffers from methods which may obscure significant results (pooling bees, genus-level classification w/Silva database). Some of the reported results are contrary to well established bee gut microbiome studies. Overall, the manuscript was very well written, easy to read and understand, but the microbiome portion would need to be re-analyzed resulting in many of the statistics and figures needing to be recreated.

We thank the reviewer for their extremely valuable comments regarding study design and data analysis, which we have given much consideration and will use to inform our future work. However, we do not feel that reanalysis of the microbiome data for the current study is warranted; please see our comments below.

Major Issues:

- I very much appreciated the authors, splitting tissues into crops, midguts, and hindgut, however I'd argue pooling is good for screening samples (toxin effects, virus titers), but less so for subtle differences in the microbiome. Because the design was such that foragers were randomly taken without accounting for age it is difficult to consider how real the differences are between healthy and stressed colonies. Like other animal systems, the honey bee gut microbiome ages as well.

We believe that the pooled samples approach was appropriate for our study, as the aim was to use the bee gut microbiome as an indicator of the overall hive microbiome. Rather than subtle differences between individual bees, we were interested in colony health and function as a whole, which is contributed to by forager bees of all ages. A single individual bee would not be representative of the whole hive, and the cost of sequencing many individual bees from the same hive would be prohibitive.

- Honey bee microbiome research has advanced far enough that we can more accurately assign ASV to the species-level. Simply, genus-level classification is not appropriate at this stage. See BEEexact for accurately classifying honey bee bacteria to species-level, which substantially improves classification over Silva database: DOI:10.1128/msystems.00082-21

Having investigated the BEEexact database after learning of it from the reviewer comments, we agree that it looks like a great resource for taxonomy assignment and we will certainly consider integrating it into our future studies. However, reanalysis of the current study using the BEEexact database at this stage would mean redoing most of the study and we do not feel that this is warranted for the following reasons:

- (1) This is a large study and involved work over a long period of time, beginning around April of 2022 when sample collection and analysis methods were put in place. This was before the BEEexact database (published in April of 2021, though earliest citations of it are from 2022) was widely accepted. Our choice of the SILVA database was based on methods used by colleagues in the field and its prominence in the literature. As it is a widely used and frequently updated database, we considered it the best one to use at the time.
- (2) Even now, the BEEexact database does not have ubiquitous use in the literature. Many other studies have come out between 2022 and now that have continued to use the SILVA database (including in high impact journals). See for example:

Nature Ecology and Evolution – Aug 2022 *

<https://www.nature.com/articles/s41559-022-01840-w>

mBio – Aug 2022

<https://journals.asm.org/doi/full/10.1128/mbio.01131-22>

Molecular Ecology – Aug 2022 *

<https://onlinelibrary.wiley.com/doi/pdf/10.1111/mec.16769>

Applied and Environmental Microbiology – Jun 2022 *

<https://journals.asm.org/doi/full/10.1128/aem.00203-22>

Scientific Reports – Nov 2022

<https://www.nature.com/articles/s41598-022-23287-6>

Journal of Invertebrate Pathology – Mar 2023

<https://www.sciencedirect.com/science/article/pii/S0022201123000265>

Journal of Invertebrate Pathology – Nov 2022

<https://www.sciencedirect.com/science/article/pii/S0048969722050409>

This list includes publications by Nancy Moran and Philipp Engel's groups (starred above), who are leaders in bee microbiome research and produce high-impact, highly cited work in this area.

- (3) The BEEexact paper states itself that taxonomy assignment between SILVA and BEEexact at the phylum to genus level are very similar and it was primarily species-level assignment where differences were noted:

“Similar performances were exhibited on a per study basis by all BEEexact and SILVA training sets when considering mean \pm SE classification rates at the phylum (97.8% \pm 0.5% versus 99.01% \pm 0.1%; $P = 0.9666$), class (97.02% \pm 0.6% versus 99.8% \pm 0.1%; $P = 0.8105$), order (94.13% \pm 1.2% versus 99.0% \pm 0.2%; $P = 0.0728$), family (92.5% \pm 1.5% versus 97.4% \pm 0.4%; $P = 0.0740$), and genus (89.2% \pm 1.8% versus 87.5% \pm 1.0%; $P = 0.9949$) levels. However, at the species level, BEEexact enabled strikingly higher classification rates compared to SILVA (81.0% \pm 1.8% versus 28.4% \pm 1.6%; $P < 0.0001$).”

We did not consider species-level assignment for the 16S data in the current manuscript. As our study design involved pooling bee samples with the intent of using the bee gut microbiome as an indicator of the overall hive microbiome, we felt that genus-level analysis would most appropriately address our aims. Therefore, it does not appear that reanalysis using the BEEexact database would be likely to produce significantly different results. For the ITS dataset, species level assignment is standard as part of the UNITE database, but as the BEEexact database does not contain fungal taxonomy this would also not affect the results of our study.

- Did the authors control for contamination in extraction and sequencing? Red flags for me were 25% Other in the Hindgut. The proportion of "Other" bacteria in each sequenced niche is inexplicable when studies cited here report that the "core" microbiome usually makes up >90% of reads. I have not

seen *Dysgonomonas* in honey bees at the levels reported here. The % relative abundance of *Hafnia* being higher than other core microbial members. For a way to remove contams from dataset See decontam: <https://github.com/benjineb/decontam> but this also might be artifacts from using Silva over BEEExact

We do not believe there has been an issue with contamination. As recommended by best practice, we used a water control and took it through the extraction and sequencing process, after which there was 574 reads compared to the average of 75005 reads per test sample. We have included this information in the supplementary methods (section entitled Amplicon Sequencing and Analysis). Please also note that the 'other' category contains taxa that were identified but present at low frequency and this is not a category denoting reads that could not be identified.

Minor Issues:

- It's important to have the databases you use in the main manuscript instead of supplement.
Added "Taxonomy was assigned using the SILVA database release 138.1 for 16S and the UNITE database release 27.10.2022 for ITS".
- Line 269: Likely *Paenibacillus alvei* unless your hives were experiencing an AFB outbreak.
Changed to: "The *Paenibacillus* genus includes *Paenibacillus larvae*, the causative agent of the fatal bee disease American Foulbrood, and *Paenibacillus alvei*, associated with European Foulbrood (32)."
- Line 285: The main *Apilactobacillus* in honey bees is *A. kunkei* which is incredibly host-adapted. These are likely not the same one's from the foraging environment.
Changed to: "The predominance of *Apilactobacillus*, a genus recently reclassified from *Lactobacillus* (35), is consistent with other studies looking at the crop (36) where the main species of *Apilactobacillus* is the host-adapted *A. kunkei*. With the microbial composition of the crop considered to be more transient and reflective of exposure to nectar and the environment, and the fact that *Apilactobacillus* also contains many species found in the sugar-rich environments of flowers and fruits, these results could be further evidence of differential foraging behaviour between bee colonies in healthy and stressed hives."
- Line 386 "For bee bread samples, 100 mg was used and for swab samples, swab heads were cut off using sterile scissors, removing any hard plastic pieces.", add a comma after "used" for improved readability
This has been corrected.
- Line 393 "Final sample numbers were n=6 for hive environment samples, n=10 for 16S bee gut samples, and n=6-8 for ITS bee gut samples." to "Final sample numbers were as follows: n=6 for hive environment samples, n=10 for 16S bee gut samples, and n=6-8 for ITS bee gut samples." This improves clarity and readability.
This has been corrected.
- Line 399: Replace "Whether data were normally distributed was determined using the Shapiro-Wilk normality test." with "The Shapiro-Wilk normality test was used to determine whether the data were normally distributed."
This has been corrected.

- In the sentence "Amplicon data (ASVs) were rarefied to even sampling depth prior to the calculation of alpha diversity metrics, or subject to centred-log ratio transformation for beta diversity measures.", change "or" to "and" to clarify that both methods were used in the analysis.

This has been corrected.

- Line 403 Kruskall-Wallis should be Kruskal-Wallis , in general fix small misspellings throughout manuscript.

This has been corrected and we have rerun a spell check to identify and correct a few other misspellings.

May 18, 2023

Prof. Dee A Carter
The University of Sydney
School of Life And Environmental Sciences
LEES Building F22, School of Life And Environmental Sciences
University of Sydney
Sydney, New South Wales 2006
Australia

Re: Spectrum00742-23R1 (Low levels of hive stress are associated with decreased honey activity and changes to the gut microbiome of resident honey bees)

Dear Prof. Dee A Carter:

Your manuscript has been accepted, and I am forwarding it to the ASM Journals Department for publication. You will be notified when your proofs are ready to be viewed. Please note that you will be expected to revise your Data Availability Statement (Line 429) to include the relevant materials and accession information.

Sincerely,

Jeffrey Gralnick
Senior Editor, Microbiology Spectrum